# WikiChurches: A Fine-Grained Dataset of Architectural Styles with Real-World Challenges

**Björn Barz**
Computer Vision Group
Friedrich Schiller University Jena
Jena, Germany
`bjoern.barz@uni-jena.de`

**Joachim Denzler**
Computer Vision Group
Friedrich Schiller University Jena
Jena, Germany
`joachim.denzler@uni-jena.de`

## Abstract

We introduce a novel dataset for architectural style classification, consisting of 9,485 images of church buildings. Both images and style labels were sourced from Wikipedia. The dataset can serve as a benchmark for various research fields, as it combines numerous real-world challenges: fine-grained distinctions between classes based on subtle visual features, a comparatively small sample size, a highly imbalanced class distribution, a high variance of viewpoints, and a hierarchical organization of labels, where only some images are labeled at the most precise level. In addition, we provide 631 bounding box annotations of characteristic visual features for 139 churches from four major categories. These annotations can, for example, be useful for research on fine-grained classification, where additional expert knowledge about distinctive object parts is often available. Images and annotations are available at `https://doi.org/10.5281/zenodo.5166986`.

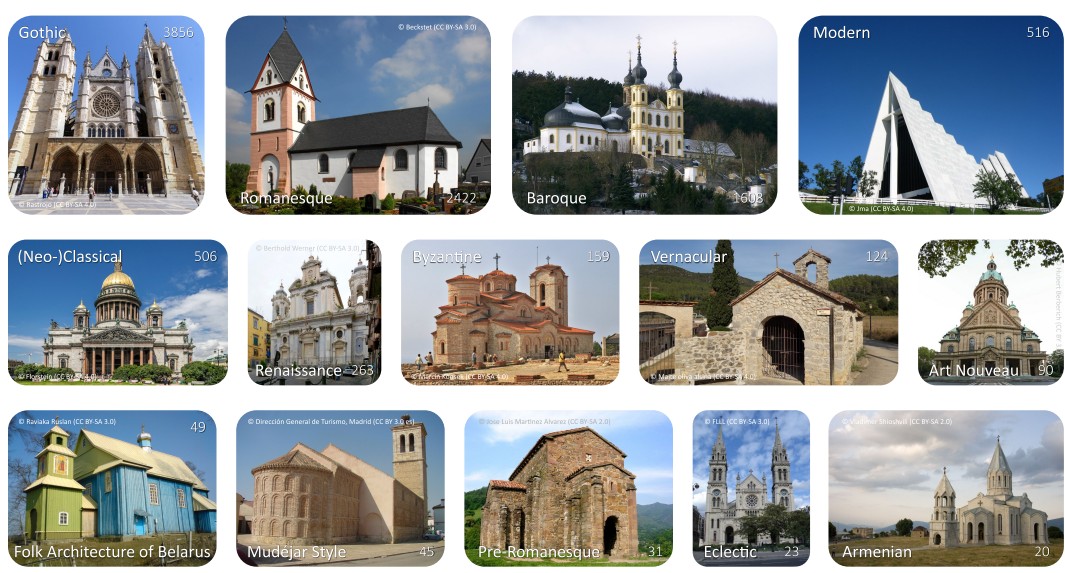

Figure 1: Example images from the 14 main styles ($\geq$ 20 images) in the WikiChurches dataset. The numbers specify the number of images in the respective category, including sub-categories.

35th Conference on Neural Information Processing Systems (NeurIPS 2021) Track on Datasets and Benchmarks.

# 1   Introduction: Why Churches?

Determining the architectural style of a building is a difficult task, even for experts, since the categorization often depends on a combination of subtle and sometimes tiny visual features such as quatrefoils or fleurons, which are both typical for Gothic architecture. The distinction between styles is furthermore a fine-grained one because architecture evolved successively. Most elements that are characteristic for Baroque architecture, for example, are based on the style of the Renaissance and simply exaggerated (i.e., higher, grander, more decorated) in the Baroque. Moreover, regional differences contribute to a high intra-class variance. English Gothic and Italian Gothic architecture, for example, look quite different but make use of the same stylistic elements. Architectural style classification is hence a challenging task even for modern computer vision methods, which often struggle with the recognition of small details [1, 2].

Among the longest standing structures are churches. For instance, the *Basilica of Santa Maria in Trastevere* in Rome was completed in 1143 and has been surviving almost nine centuries. The full variety of architecture from medieval to modern times is hence reflected in church buildings, making them an obvious choice for the study of architectural styles.

In this work, we present *WikiChurches*, a dataset comprising 9,485 images of 9,346 church buildings in Europe, labeled with their architectural styles. The meta-data about the churches and corresponding images were extracted from Wikidata [3], a community-driven knowledge base striving for organizing the knowledge contained in Wikipedia in a structured form as knowledge graph.

The WikiChurches dataset poses numerous challenges that are frequently encountered in real-world machine learning applications:

- a **small sample size** compared to popular large-scale deep learning benchmarks with millions of images [4–6], which is a common problem in practice when data acquisition or annotation is expensive, requiring data-efficient deep learning methods [7, 8],

- a **highly imbalanced class distribution**, where machine learning methods are prone to focusing on the larger classes but performing substantially worse on underrepresented ones,

- **fine-grained distinctions** between classes based on small and subtle features, which is an important problem with a very active research community and established regular competitions [9, 10],

- a **high variance of viewpoints**, requiring the robust recognition of distinctive structures under strong deformations,

- a **label noise** of approximately 6.5%, which is a common problem with web-sourced or crowd-sourced data and calls for outlier-robust learning techniques,

- a **hierarchical structure of labels** (e.g., *Perpendicular Style* is *English Gothic* is *Gothic*), which opens up potential benefits through the use of hierarchical classification methods [11, 12] but also additional challenges, since only few samples are labeled at the most precise level of the hierarchy.

These challenges render the WikiChurches dataset useful as a benchmark and playground for various research areas, including fine-grained visual recognition [9, 10], data-efficient deep learning [7], dealing with imbalanced training sets, and hierarchical classification with imprecise labels [13].

In addition to the class labels obtained from Wikidata, we provide 631 bounding box annotations for 139 selected churches from four major categories. These bounding boxes hint to important structural features that are particularly characteristic for a certain style, e.g., the aforementioned quatrefoils. Techniques that utilize this additional information to increase classification accuracy would also be of interest in other domains where small characteristic object parts are crucial for making a correct decision, e.g., the classification of bird species [14–16]. Another research area that could make use of these annotations is interactive image retrieval, where the user highlights particularly relevant parts of query images to refine the search [17, 18]. WikiChurches can be used as a testbench for such a setting by simulating user feedback using the provided bounding box annotations.

The remainder of this paper is organized as follows: In Section 2, we describe how the data was collected, cleaned, and pre-processed, as well as how the additional bounding box annotations were obtained. The resulting WikiChurches dataset is then presented in Section 3. We also conduct a baseline classification experiment in Section 4 to provide an impression of the difficulty of the dataset. Related work is covered in Section 5 and Section 6 concludes this work.

## 2 Dataset Creation

Because churches are often striking structures and considered as touristic sights, Wikipedia features many articles on such individual buildings, typically mentioning their history and architectural style as well as providing images of the church. Capturing such factual information provided in the text in a machine-readable format like a knowledge graph is the goal of the Wikidata project [3], which hence is an ideal source for collecting information about churches.

### 2.1 Data Collection

Wikidata enabled us to quickly and reliably obtain a list of churches with style labels and images by querying its SPARQL [19] endpoint. Specifically, we programmatically request a list of all churches linked with an architectural style, a country, and at least one image on Wikimedia Commons. We furthermore query the geographical coordinates and year of construction if available.

The country is not optional, since we use it for restricting the search to buildings in Europe. There are two main reasons for this decision: First, Europe has the longest history of Christianity and hence the broadest historical variety of church buildings. Second, the domain experts we consulted regarding additional annotations were most versed in the field of European architecture.

The exact SPARQL query we used can be found in the appendix. The query was executed on July 12th, 2017, and resulted in a set of 10,625 churches.

We then downloaded the images linked to each church on Wikidata, converted them to JPEG, and resized larger images so that their smaller side has a maximum size of 1280 pixels. Meta-data such as original URL, author, and license are provided in a separate central file. All images on Wikimedia Commons are under a permissive license allowing redistribution.

### 2.2 Data Cleaning

Not all images linked to a church on Wikidata are actual photographs of that church building. Approximately 3% of the images downloaded by us showed the interior instead of the exterior of the church. We identified these images with the assistance of a pre-trained indoor-outdoor classifier [20], ranking all images by decreasing score for the prediction "indoor". The images in this ranked list were verified as indoor images manually until the results of the classifier became reliable enough, so that we did not expect a significant amount of further indoor images in the rest of the ranked list.

In a second step, we browsed through all remaining images and removed those that were not photographs of the exterior of a church building. This includes close-ups of individual objects (wall sculptures, pictures, organs etc.), scans of historical drawings, city scenes where the church is just one of many buildings, heavily truncated and occluded buildings, and ruins of former churches.

These two cleaning steps resulted in the removal of over a thousand images and led to the final dataset of 9,485 images. Churches for which not a single image remained were removed from the dataset. The metadata associated with churches for which not a single image remained was removed from the dataset as well.

### 2.3 Class Hierarchy Construction

Each church we queried from Wikidata is linked to one or more architectural styles, but styles can themselves also be linked to more general super-classes. For example, the *Perpendicular Style* is a type of *English Gothic Architecture*, which itself is an instance of *Gothic Architecture*. For all styles in our dataset, we recursively queried their parent styles from Wikidata to obtain the class hierarchy.

We made a few modifications to the style taxonomy obtained this way from Wikidata. To begin with, we made *Rococo* a sub-category of *Baroque*, since it denotes the last period of the Baroque epoch and is sometimes also referred to as *Late Baroque*. However, this link did not exist in Wikidata.

The remaining changes all concern the super-class *Historicism* and the various styles contained therein. Historicist buildings imitate a historical style but have been constructed at a later time. For example, the *Église Saint-Pierre* in Rezé, France, uses stylistic elements of Gothic architecture but has been built in 1867, three centuries after the end of the Gothic epoch. This out-of-period construction

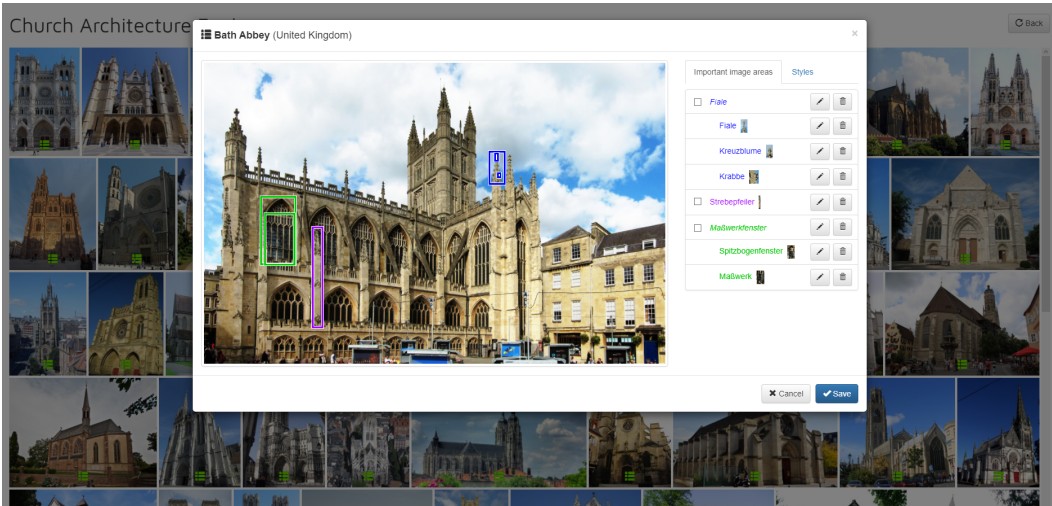

Figure 2: Screenshot of the web-based annotation interface provided to the domain expert for creating bounding box annotations of characteristic structures.

is the defining commonality among all subclasses of Historicism, while they have not much in common visually. Distinguishing these buildings from the style they imitate, however, is often difficult. Sometimes, the use of materials or construction techniques that were not available at the original time can provide useful clues, but often the year of construction is the only reliable indicator. For a computer vision dataset, it is hence more reasonable to organize the several Historicism classes as sub-categories of the style they imitate, e.g., *Gothic Revival* as a sub-class of *Gothic*.

## 2.4 Additional Annotations

To augment the dataset with additional information, we asked a domain expert from art history to provide bounding box annotations indicating distinctive structures that are characteristic for certain architectural styles. For this step, we restricted the set of styles in question to four well-studied ones, which account for a large fraction of the dataset: *Romanesque*, *Gothic*, *Renaissance*, and *Baroque*.

For annotating the images, we provided a web interface that allowed browsing all images per category, including sub-categories. The expert was asked to focus on images of churches which can be considered pure in style and prototypical for that period. Bounding box annotations could then be created for each selected image in a popup window by dragging the mouse over a larger version of the image (see Fig. 2. Each individual bounding box was labeled with a description of the characteristic structure it contains. Boxes could furthermore be grouped to indicate that several individual elements form a larger structure of interest.

## 3 The WikiChurches Dataset

In the following, we present the WikiChurches dataset in detail. We also propose various subsets of WikiChurches that could be of use for different research fields and investigate issues such as label noise and biases. Example images from the 14 largest categories of WikiChurches are shown in Fig. 1. The dataset can be obtained at `https://doi.org/10.5281/zenodo.5166986`.

## 3.1 Churches and Images

The full WikiChurches dataset comprises 9,485 images of 9,346 different church buildings. This difference indicates that multiple images are available for some churches. However, this is only the case for 1.2% of all churches in the dataset (see Fig. 3b). One particular church, the *Village Church of Berlin-Rahnsdorf* in Germany, is represented with the maximum of six images.

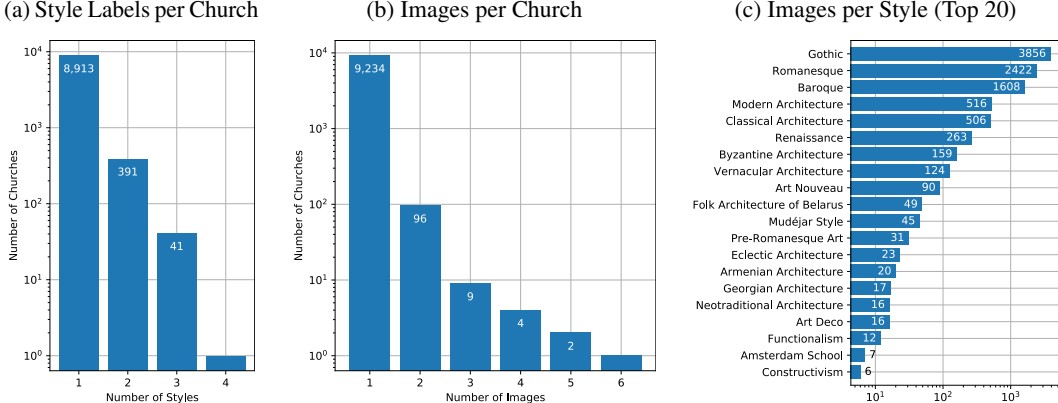

Figure 3: Statistics of the full WikiChurches dataset. Note that all histograms are on a log-scale.

## 3.2 One Church, Many Styles

Churches can not only have multiple images but also more than one style label. This is due to the long history of these buildings, which have repeatedly been renovated, extended, and altered. For example, *Roskilde Cathedral* in Denmark was the prototype of the then novel *Brick Gothic* architecture. Constructed in the 12th and 13th century, it also incorporates elements of the previously dominating *Romanesque* style. Because it served as the main burial site for Danish monarchs over centuries, there has been frequent need for additional chapels, which were built in the style that was currently in vogue during that time. Therefore, parts of the cathedral do not conform to the original architecture but exhibit the styles of the *Renaissance*, *Neoclassicism*, *Byzantine Revival*, and even *Modern* architecture. In Wikidata, however, this cathedral is only labeled with its dominant style, Brick Gothic. We expect this to be the case for many churches that exhibit a mixture of styles. Thus, it is important to keep in mind that labels in WikiChurches are not mutually exclusive. Most churches are labeled with their dominant architecture but can incorporate additional styles. Techniques such as label smoothing [21] could prove useful in this situation.

The task of architectural style recognition is hence actually a multi-label problem. However, only 4.6% of the churches in WikiChurches have more than one label (see Fig. 3a). The maximum of four style labels is obtained by the *All Saints Church* in Shirburn, UK, while one of these labels (*Gothic*) is a super-class of another label (*English Gothic*) and hence redundant. Because the multi-label nature of architectural styles is underrepresented in WikiChurches, we restrict our canonical subsets introduced in Section 3.5 to churches with a single label, casting the problem as a single-label classification task asking for the dominant style.

## 3.3 Imbalanced Class Distribution

The buildings included in WikiChurches exhibit a variety of 117 unique style labels. Even when only considering the first level of the class hierarchy, there are still 64 different styles. Almost half of them (31), however, only comprise a single image. In general, the class distribution is extremely imbalanced and long-tailed (see Fig. 3c). The two largest classes, *Gothic* and *Romanesque*, account for two thirds of the entire dataset. Four classes already cover 88.6% of all images.

## 3.4 Class Hierarchy

The class hierarchy obtained from Wikidata is three levels deep. There are 64 architectural styles on the first level of the hierarchy, 45 on the second, and 8 on the third. Only 5 of the 45 second-level classes have further sub-classes.

Not all churches in the dataset are labeled with the highest possible precision. Many churches in *Early English Gothic* style, for instance, will simply be labeled as *Gothic*. For this reason, the majority of 5,678 churches is labeled at the first level of the taxonomy, 3,154 churches at the second, and only 81 churches at the third level. Only churches with a single label were considered for this counting.

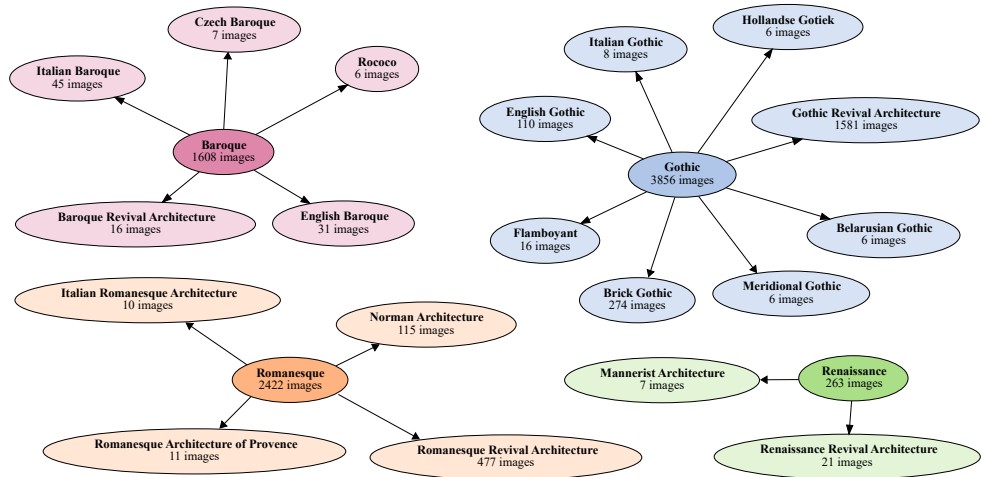

Figure 4: A subset of the WikiChurches hierarchy covering all classes in the WikiChurches-H subset.

Table 1: Key figures of the canonical WikiChurch subsets we propose.

|  | Images | Classes | Images/Class | Hierarchy Level | Bounding Boxes |
|---|---|---|---|---|---|
| WikiChurches | 9,485 | 64 – 117 | 1 – 3,856 | $1^{st}$ – $3^{rd}$ | ✗ |
| WikiChurches-14 | 8,931 | 14 | 20 – 3,856 | $1^{st}$ | ✗ |
| WikiChurches-6 | 8,488 | 6 | 264 – 3,856 | $1^{st}$ | ✗ |
| WikiChurches-4 | 7,526 | 4 | 264 – 3,856 | $1^{st}$ | ✓ |
| WikiChurches-H | 2,753 | 19 | 6 – 1,581 | $2^{nd}$ | ✗ |

A diagram depicting the full class hierarchy can be found in the appendix. Figure 4 shows the simplified two-level hierarchy of the 19 classes included in the WikiChurches-H subset (see below). Note that the numbers of images in the $2^{nd}$-level categories do not sum up to the number of images in their super-class, since the $1^{st}$-level categories contain additional images with less precise labels. Specialized techniques for learning from imprecise labels could be used to facilitate this additional information and learn better feature representations for sub-classes as well [13, 22].

## 3.5 Canonical Subsets

Due to the long-tail class distribution of the full WikiChurches dataset and the varying precision of style labels regarding their level in the taxonomy, we propose several subsets of WikiChurches that alleviate these issues. The subsets have different characteristics and could be useful for different research questions. The key facts for each subset are listed in Table 1.

**WikiChurches-14**  comprises all 14 classes from the $1^{st}$ level of the hierarchy that contain at least 20 images. Churches with more precise labels were re-labeled to their superclass (e.g, *English Gothic* to *Gothic*). Regarding the number of images, this subset still covers 94% of WikiChurches. The class imbalance is still challenging but not impossible to handle (as opposed to the case with only one image per class).

**WikiChurches-6**  is a subset of WikiChurches-14 restricted to the 6 largest classes comprising more than 200 images. These few classes account for 89% of all images in WikiChurches.

**WikiChurches-4**  is a subset of WikiChurches-6 limited to the four classes for which we provide bounding box annotations of characteristic visual features (see Sections 2.4 and 3.6): *Romanesque*, *Gothic*, *Renaissance*, and *Baroque*.

**WikiChurches-H**  is intended for studying hierarchical classification. It spans the 19 sub-classes of the four styles from WikiChurches-4 that contain more than 5 images. Churches with $3^{rd}$-level labels were re-labeled to their $2^{nd}$-level superclass. Additional images from WikiChurches-4 with less precise labels could be incorporated to learn better representations for the $1^{st}$-level classes or even all categories [13]. The class hierarchy of WikiChurches-H is depicted in Fig. 4.

(a) Gothic

(b) Romanesque

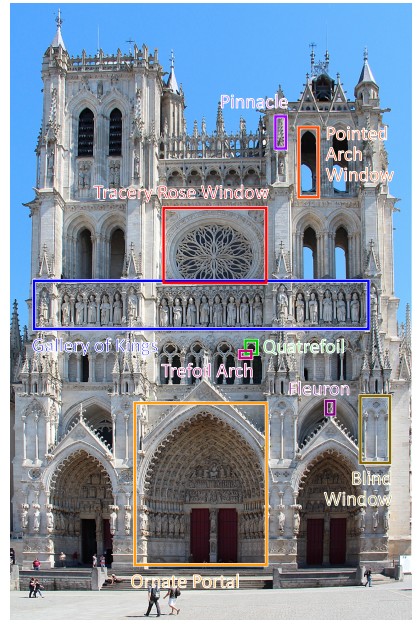

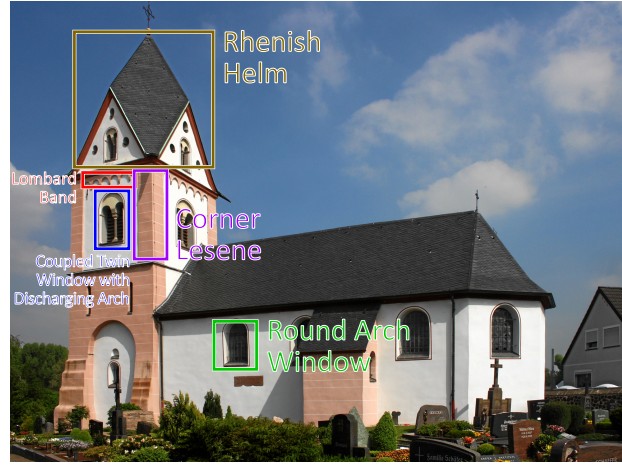

Photo: Beckstet (CC BY-SA 3.0)

(c) Details

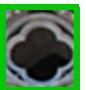 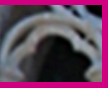 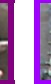 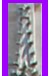 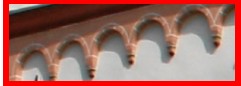

Photo: Jean-Pol Grandmont (CC BY 3.0)

Figure 5: Examples of the bounding box annotations highlighting distinctive visual features.

For WikiChurches-6 and WikiChurches-4, we provide canonical training/validation/test splits. The test split includes about 25% of the churches from each class but at least 100 and at most 500. The validation split is balanced and includes 50 churches from each class. All remaining images are used for the training split. We do not provide canonical splits for the other two subsets since we cannot anticipate the specific needs of the research questions for whose study they could be used.

Table 2: Bounding box statistics.

| Style | Images | Boxes |
|---|---|---|
| Romanesque | 54 | 235 |
| Gothic | 49 | 290 |
| Renaissance | 22 | 93 |
| Baroque | 17 | 31 |

## 3.6 Bounding Box Annotations of Characteristic Structures

WikiChurches provides 631 bounding box annotations of distinctive visual features for 139 selected images from the four categories in WikiChurches-4. Per-class statistics can be found in Table 2. On average, there are 4.5 bounding boxes per annotated image. The highest number of bounding boxes on a single image is 10 and obtained by three images.

Bounding boxes can also be organized in groups to indicate that they form a larger structure. While most of the 566 groups in the dataset contain only a single bounding box, 11% are actual groups. No group has more than three elements.

Figure 5 shows two examples of annotated images from the dataset. The labels of the annotated architectural structures can be quite detailed, such as "coupled twin window with discharging arch", and are hierarchically organized. The aforementioned label, for example, would be a hyponym of both "coupled twin window" and "discharging arch". Table 3 shows a list of the most frequently annotated architectural elements.

Table 3: Most frequent box labels.

| Bounding Box Label | Boxes |
|---|---|
| Tracery | 53 |
| Round Arch Window | 41 |
| Pointed Arch Window | 37 |
| Buttress | 31 |
| Pinnacle | 28 |
| Lombard Band | 25 |
| Pilaster | 20 |

The size of the annotated structures in relation to the entire building varies widely. Some important structures are particularly small, such as quatrefoils, fleurons, and pinnacles, which requires processing the images at a high resolution.

### 3.7 Label Noise

As opposed to the bounding boxes provided by the domain expert, the style labels in the WikiChurches dataset have been contributed by the Wikipedia community and can hence be expected to be more noisy. To quantify the label noise in the dataset, we asked an expert to inspect a random subset of 200 single-label images from WikiChurches and check the correctness of their labels. Labels could be denoted as either correct, partially correct, or wrong. A label is considered partially correct if the image exhibits not only the labeled style but also other prominent styles. Additionally, the expert could denote an image as uninformative if it did not show enough details of the church to determine its architectural style. Images that received this verdict typically show only parts of the building, e.g., a tower, or a side of the building that does not exhibit enough characteristic features.

93.5% of the inspected images were at least partially correct (86.1% were completely correct), 4.0% were incorrectly labeled and 2.5% were uninformative. Label noise hence is in fact an issue with this dataset as with any web-sourced dataset. However, the amount of noise is within a reasonable range that can also occur in practice for data as difficult to annotate as architectural styles. Just checking the correctness of a given label took our domain expert two minutes per image on average.

### 3.8 Biases

By design, the WikiChurches dataset only covers European architecture and, in particular, the architecture of churches. It is mainly intended to be used as an interesting benchmark for computer vision methods and not as a representative and comprehensive source of training data for architectural style classifiers. While the latter might work well for common styles in Europe, it will not generalize to other cultures and maybe not even to buildings different from churches. In the worst case, a too strong focus on techniques for WikiChurches bears the risk of marginalizing other cultures.

Besides the intended cultural and geographical bias towards Europe, the geographical distribution of churches within Europe is highly imbalanced as well. As shown in Fig. 6, half of all churches in the dataset are from Germany and France (38% from Germany, 11% from France). Two thirds of the dataset concentrate on as few as four countries: Germany, France, the UK, and Spain. This distribution is not representative of the actual distribution of church buildings across Europe but most likely correlated with the size and level of activity of the local Wikipedia communities and their propensity to enter information in Wikidata.

Similarly, the distribution of styles in the dataset (see Fig. 3c) is not representative of reality either. That *Gothic* and *Romanesque* architecture account for the largest portion of the dataset does not imply that they are the most common styles of churches in Europe but simply popular among photographers.

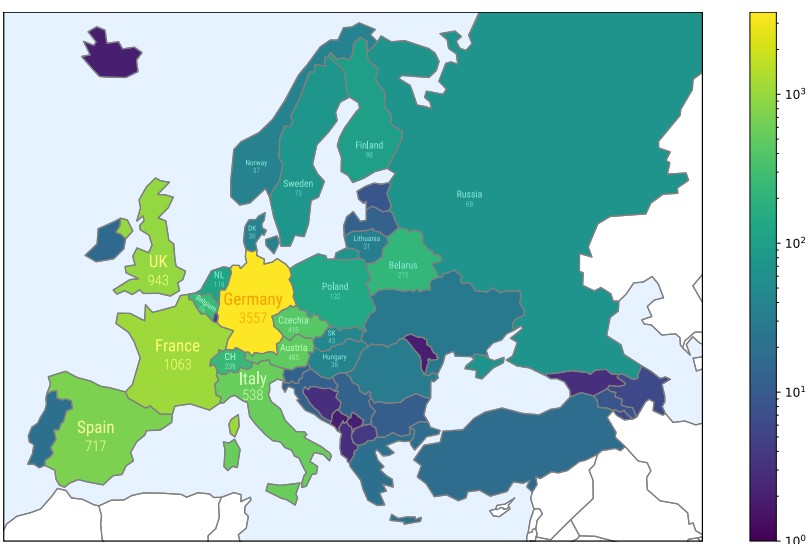

Figure 6: Heatmap showing the geographical distribution of churches in WikiChurches per country. Note that colors are scaled on a log-scale. Exact numbers are provided for the top 20 countries.

# 4 Classification Baseline

We conduct a simple baseline experiment on the WikiChurches-6 subset for assessing its difficulty.

## 4.1 Setup

We use a ResNet-50 architecture [23] for our baseline classifier, pre-trained on ImageNet-1k [24] and fine-tuned for 80 epochs using SGD with a cosine annealing learning rate schedule [25]. The initial learning rate and the weight decay factor are optimized on the training and validation split using Asynchronous Hyperband with Successive Halving [31]. The final classifier is then trained on the combined training and validation split and evaluated on the test split. The used hyper-parameters can be found in the appendix.

Regarding data augmentation, we extract a random crop from the original image, whose area accounts for between 40% and 100% of the image and whose aspect ratio is sampled from the interval $[\frac{3}{4}, \frac{4}{3}]$. This crop is then resized to $224 \times 224$ pixels and flipped horizontally in 50% of the cases. During inference, we resize the image so that its smaller side is 224 pixels and extract a square center crop.

We furthermore investigate two variants of the basic setup: First, we account for the highly imbalanced distribution of training classes by randomly sub-sampling the training dataset. For each epoch, a subset of samples is drawn from each class without replacement, whose size matches that of the smallest class. Second, we double the input resolution from $224 \times 224$ to $448 \times 448$ pixels.

In addition to this baseline, we evaluate fine-tuning of a TResNet-M architecture pre-trained on the larger and more diverse ImageNet-21k dataset using a semantic multi-label training scheme based on the WordNet hierarchy of classes [26]. This technique is the current state of the art on the Stanford Cars dataset [27], according to *Papers With Code*[1]. Stanford Cars comprises images of different car models and is similar to WikiChurches in the regard that it is small-scale, fine-grained, and its topic should be well covered by ImageNet. Fine-tuning this pre-trained network can hence be expected to provide good performance on WikiChurches as well.

Since the classes are not uniformly distributed in the test split of WikiChurches-6, we measure the classification performance in terms of balanced accuracy, i.e., the average of all per-class accuracies. We repeat each experiment 10 times and report the average accuracy along with its standard deviation.

## 4.2 Results

Balanced accuracies and confusion matrices resulting from our experiments are shown in Fig. 7. The low-resolution baseline fine-tuned from ImageNet-1k pre-training with imbalanced classes achieves an accuracy of 60.2%, which is much better than a constant prediction (which would achieve 16.7%) but still rather poor for a six-class problem. Sub-sampling the classes during training to a uniform distribution leads to a substantial drop of accuracy by 11 percent points for the majority class but improves the accuracy for all other classes, resulting in an overall improvement of 1.5 percent points.

Additionally doubling the input resolution increases the accuracy further by 4.2 percent points. Most of all, the higher resolution facilitates recognizing *Renaissance* architecture (35% improvement) and distinguishing *Gothic* from *Romanesque* architecture, whose direct successor it is (11% improvement).

Interestingly, *Modern* architecture is most accurately recognized, even though up to seven times more training data is available for other classes such as *Gothic*. Despite the high variance of shapes of modern buildings, this class might be easily recognized because it is most different from all others. *Gothic*, *Romanesque*, and *Baroque* architecture obtain reasonable accuracies most likely because a large amount of images is available for these classes. *Renaissance* is the most difficult category, not only because it comprises the smallest amount of training images but also because it exhibits similar visual features as *Baroque* architecture, just not as exaggerated and ornamented. Thus, as much as 36% of images from this class are erroneously classified as *Baroque*.

The TResNet-M pre-trained on the larger and more diverse ImageNet-21k mainly improves the performance on this most difficult class, resulting in a total balanced accuracy of 69.5%, an improvement of 3.6 percent points over the ImageNet-1k pre-training. The increase is similar for lower-resolution

---

[1] https://paperswithcode.com/sota/fine-grained-image-classification-on-stanford?p=imagenet-21k-pretraining-for-the-masses, retrieved: October 17th, 2021.

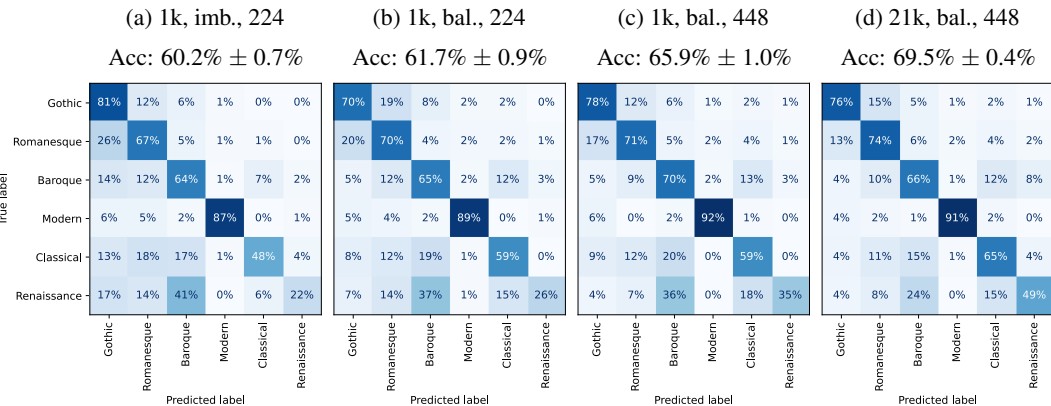

Figure 7: Confusion matrices for four variants of our baseline classifier on WikiChurches-6. Sub-figure captions specify the pre-training dataset (ImageNet-1k or ImageNet-21k), whether classes were balanced (bal.) during training or not (imb.), and the input image resolution. Classes are sorted by decreasing number of samples. Accuracy is measured as mean balanced accuracy over 10 runs. The confusion matrices show the performance of the best model among all runs.

input images (4.8 percent points) but less pronounced for training with imbalanced classes (1.4 percent points). This approach still misclassifies over one third of the images for half of the classes in WikiChurches-6, emphasizing the difficulty of architectural style classification.

## 5   Related Work

There are a few other datasets for architectural style classification. Most related to our data collection approach is that of Xu et al. [28], who also obtain images from Wikimedia Commons. Instead of using Wikidata as a source for labels, they download all images from a pre-defined set of 25 Wikimedia Commons categories corresponding to architectural styles. Nested categories allow them to obtain a class hierarchy as well. In contrast to them, our WikiChurches dataset provides twice as many images and furthermore bounding box annotations for characteristic architectural elements.

The Architectural Heritage Elements image dataset (AHE) [29] focuses exclusively on such distinctive structures and comprises 10,235 images from 10 categories of architectural style elements such as domes, columns, gargoyles etc. However, AHE does not contain images of entire buildings and no labels of architectural style, which makes it unsuitable for architectural style recognition.

The only other dataset besides ours that combines both types of annotations is MonuMAI [30]. All 1,500 images in this dataset are annotated with both their dominant architectural style and bounding boxes around distinctive elements. Architectural styles are limited to four categories and bounding box labels to 15 key elements. With 6,600 bounding boxes in total, the number of such annotations is ten times higher than in WikiChurches. Our dataset, in contrast, provides six times more images and covers much more architectural styles. With 92 unique bounding box labels, we furthermore provide a larger variety of architectural elements than the 15 categories in MonuMAI.

## 6   Conclusions

We presented the WikiChurches dataset for architectural style classification, consisting of images of churches from Wikimedia Commons and hierarchical style labels from Wikidata. In addition to these community-provided labels, we provide complex bounding box annotations for characteristic visual features for four selected styles. The dataset poses many real-world challenges such as an imbalanced class distribution, fine-grained and subtle distinctions between classes, small but important visual features, a small sample size, and a hierarchical organization of labels. Baseline classification experiments demonstrate the difficulty of the task, achieving a non-trivial but still highly unsatisfactory accuracy of 69.5% with as few as six classes. We hope that WikiChurches with its various interesting properties will be of use for a broad range of research areas.

## Acknowledgments and Disclosure of Funding

We would like to express our gratitude to numerous people who were involved in the creation of this dataset: Katharina Saul (University of Marburg, Department of Arts) provided the excellent annotations of distinctive visual features. Her expertise regarding architectural styles furthermore contributed to shaping the dataset into its current form. Marija Salvai (independent architect) helped us with assessing the amount of label noise in the dataset. Finally, we would like to thank Prof. Dr. Thomas Erne (University of Marburg, Institute for Church Architecture) for the inspiring discussions that sparked the idea for creating this dataset.

This work was supported by the German Research Foundation as part of the priority programme "Volunteered Geographic Information: Interpretation, Visualisation and Social Computing" (SPP 1894, contract number DE 735/11-1).

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
