# OpenReview forum: "WikiChurches: A Fine-Grained Dataset of Architectural Styles with Real-World Challenges"
_NeurIPS.cc/2021/Track/Datasets_and_Benchmarks/Round2 — NeurIPS 2021 Datasets and Benchmarks Track (Round 2)_

### Official Review · Reviewer_v7xN · 2021-09-05
**Interesting multi-use dataset**

**Rating:** 6
**Confidence:** 4
**Correctness:** Yes.

**Strengths:**

The paper is well written. The dataset provides a large, comprehensive collection of images that is validated by a known expert on church architecture in Europe. The architectural styles are well curated and some images contain relevant bounding box information that could be used for tasks that require additional characteristic object annotation.

The model exploration, while not novel, is extensive in its exploration of class balancing issues, resolution adjustment and explanation for the difficulty of classifying the images.


**Weaknesses:**

The dataset includes a hierarchical organization of labels, where only some images are labeled at the most precise level. The majority of 5678 churches are labeled only at the first level of the taxonomy, while only 81 churches are labelled at the most precise third level. The labels are hugely imprecise, and would limit many works tackling hierarchical classification.

In addition, the authors acknowledged that the task of architectural style recognition is actually a multi-label problem. Casting the task to just the dominant style adds an additional layer of uncertainty in the classification task. It’s not clear whether visual features are indicative of class label, or just the time period. Without further historical context, could humans learn to classify this?

The bounding box labels are unique, but not all objects of interest are labelled in a single image. This seems to be in contrast to MonuMAI [29], where every object is labelled, and has more flexibility for different vision tasks.

The authors mention that Renaissance is a notoriously difficult category to recognize since it has few images and overlapping architectures. Given the huge accuracy disparity for this category versus others, there is potential for this category to be augmented.

Regarding the label noise, was the random selected images subset a uniform sampling or sampled per category? How were the number of images determined, since there might be a larger sample size for a certain desired confidence interval? If an image was labeled as uninformative, why was this label provided and could another expert correctly annotate it? How well could the baseline model classify these incorrectly labeled or uninformative images?


**Additional Feedback:**

See above.

**Clarity:**

Paper is well written and easy to follow.


**Documentation:**

Maintenance plan to be discussed.


**Ethics:**

No.

**Relation To Prior Work:**

Prior work is discussed. Authors should additionally clarify how each of the canonical subsets they propose compare to prior work, with respect to attributes such as image count and architectural style count.


**Summary And Contributions:**

In this paper, the authors introduce a dataset for architectural style classification sourced from Wikipedia. They additionally provide 631 bounding box annotations for 139 selected churches from four major categories. The dataset tackles challenges with small sample size, highly imbalanced class distribution, fine-grained distinctions, high variance of viewpoints, label noise, hierarchical structure of labels. The paper formalizes the task of architectural classification and its corresponding difficulties.

---

> ### Author Response · Authors · 2021-09-29
> **Response**
>
> Thank you for your thoughtful comments. We would like to address your questions regarding the label noise study, the necessity of historical context, and the utility for hierarchical classification in the following.
>
> ### Hierarchical Classification
>
> You pointed out that only 81 churches are annotated at the most precise 3rd level of the class hierarchy in the raw form of the dataset. We would like to note that a much larger number of 3,154 of the 8,913 churches are labelled on the second level and that only a few second-level classes have a third level of categorization at all (see Fig. 8 in the appendix). Thus, the third level is not always "the most precise" one.
>
> For this reason, we designed the WikiChurches-H variant, which is restricted to two levels of the taxonomy. Images with 3rd-level labels where re-annotated to their 2nd-level super-class. This variant of the dataset comprises 2,753 images labeled on the second level of the hierarchy from the four super-classes Baroque, Gothic, Romanesque, and Renaissance.
>
> Additional 4,773 imprecise labels for these four first-level classes are available in WikiChurches and can be used for learning with imprecise labels (e.g., [[13]][1]). This leads to a dataset with 7,526 images, of which 2,753 are labeled at the more precise hierarchy level.
>
> ### Historical Context
>
> The question whether some of the images require external data such as the year of construction to be classified correctly is an interesting one. We discussed this extensively with the domain expert from art history, who was in charge of creating the bounding box annotations.
>
> The conclusion of this discussion was that most architectural styles can in fact be recognized solely by characteristic visual features. There exist systematic handbooks and field guides enumerating these features for each architectural style, such as the [*Dictionary of Architecture and Landscape Architecture* by James Stevens Curl and Susan Wilson][2] or *La Caractéristique des styles* by Robert Ducher.
>
> The only class of styles that sometimes cannot be recognized from images only is the *Historicism* category. Styles in this category, such as *Gothic Revival Architecture* or *Neo-Byzantine Architecture*, imitate older styles but were built in later periods. In a few cases, this can be recognized by the use of modern materials or construction techniques but such clues are very subtle. Therefore, we distributed all images from sub-classes of *Historicism* to the class of the style they imitate when constructing WikiChurches. For example, the super-class of the category *Gothic Revival Architecture* has been changed from *Historicism* to *Gothic Architecture*.
>
> This is discussed in more detail section 2.3 in the paper.
>
> ### Label Noise Study
>
> *Q: Was the random selected images subset a uniform sampling or sampled per category?*
> A: We sampled the images uniformly from the dataset without considering their labels.
>
> *Q: How were the number of images determined, since there might be a larger sample size for a certain desired confidence interval?*
> A: The main factor driving our decision about the sample size for the label noise study was the time that the domain expert was able to spend on this task. We discussed this with them upfront and they agreed to revisit about 100 images but not significantly more.
>
> *Q: If an image was labeled as uninformative, why was this label provided and could another expert correctly annotate it?*
> A: We asked the domain expert to label an image as uninformative if it "does not show enough details of the church to determine its architectural style." The five images that received this verdict often show only parts of the building, e.g., only a tower, or a side of the building that does not exhibit enough characteristic features for making a confident decision.
> The following links show two examples of images that were denoted as "uninformative":
> https://commons.wikimedia.org/wiki/File:Köln_Kirche_St._Hubertus_Turm.jpg
> https://commons.wikimedia.org/wiki/File:OldChurchBorssum.JPG
> While these images allow for a *guess* of their architectural style (Modern and Romanesque in these cases), the evidence is insufficient for a definitive decision. Of course, whether the evidence is sufficient or not depends on the opinion of the expert and a different expert might come to a different conclusion. Unfortunately, however, our access to domain experts is limited.
>
> *Q: How well could the baseline model classify these incorrectly labeled or uninformative images?*
> A: Only one of the five images denoted as uninformative are contained in our test set. This image, which is the first one linked to above, could correctly be classified by our baseline model.
>
> [1]: https://ieeexplore.ieee.org/abstract/document/9413283
> [2]: https://doi.org/10.1093/acref/9780199674985.001.0001

---

> > ### Comment · Reviewer_v7xN · 2021-09-29
> > **Response**
> >
> > I am grateful to the authors for the clarifications, and am raising my score.

---

### Official Review · Reviewer_qGhq · 2021-09-16
**A dataset of architectural styles of churches;**

**Rating:** 8
**Confidence:** 4

**Strengths:**

The authors make well-motivated decisions for the design of the dataset which are described in detail in the paper. The dataset covers different types of images and labels compared to prior datasets. Hence is likely be of value to the community interested in designing computer vision techniques for analyzing architectural styles.

The discussion of label noise and biases in the dataset will allow users of the dataset to scope the utility of the models trained on this dataset (e.g., geographical bias).

The baselines are sensible and the low accuracy on this task might motivate novel computer vision and machine learning algorithms.

**Weaknesses:**

The utility of bounding boxes was not analyzed. For example, could the classifiers be improved by looking at regions within these boxes? Are existing detectors able to localize these parts from the limited amount of training data available?

**Additional Feedback:**

None.

**Clarity:**

Yes the presentation of the paper is clear. The design of the dataset and baseline experiments are well motivated.

**Correctness:**

The claims are correct. The dataset construction appears to be sound and well motivated. The baseline experiments are based on the standard practice of fine-tuning ImageNet pre-trained classifiers.

**Documentation:**

Yes, the authors have provided links to download the dataset and have posted them on zenodo. A datasheet in included and shared under a "Creative Commons Attribution-ShareAlike 4.0 International Public License" (described in the LICENSE file). The authors have also shared the weights for their pre-trained models and standardized train/val/test splits.


**Ethics:**

No.

**Relation To Prior Work:**

The contribution is primarily a dataset. The authors discussion prior datasets and work on architectural style classification from the computer vision / machine learning community.

**Summary And Contributions:**

A dataset of consisting of ~10k images of Churches labeled with architectural style is presented. This is a challenging task requiring identification of detailed visual features and understanding the relation between them. At the same time the dataset presents challenges to existing recognition systems due to the small sample sizes, data imbalance, and viewpoint variability.

The authors source the images from Wikipedia and filter images such that they depict the main building. The architectural style labels were constructed in consultation with architects and art historians, and thus automatic techniques to detect these styles in real images might enable large-scale studies in these domains. The authors present a detailed analysis of various design decisions that went the structure of the label space, dataset collection and post-processing. Several versions of the dataset are presented that vary depending on the number of class labels. Additionally, the authors identify a set of distinctive structures for which bounding-box annotations are provided.

The authors also present several baseline experiments using a ResNet50 fine-tuned on this dataset. The classification accuracy is rather low for this problem, and the authors find that the increasing image resolution and class-balancing helps.

---

> ### Author Response · Authors · 2021-09-29
> **Thank you**
>
> Thank you for your review. We are glad to hear an independent confirmation of the utility of the dataset and the discussion in the paper.
>
> An empirical investigation of the utility of the bounding box annotations is definitely something we would have liked to do. However, it would have required methodical contributions such as steering the classifier towards the distinctive features during training or a two-stage approach of detection and classification, as you suggested. We considered this out of scope for this paper, whose clear focus is on describing the dataset and its collection process. We understood this would fit the NeurIPS Datasets track well.

---

> > ### Comment · Reviewer_qGhq · 2021-09-30
> > **Use of bounding boxes**
> >
> > I agree with your thought to focus on the dataset collection.
> >
> > Another possibility is to see if the regions the are considered important for the classification task align with the annotated parts. For example you might correlate saliency maps from the classifier with parts to see which ones are the most predictive. You might even discover accidental biases in the dataset.

---

> > > ### Author Response · Authors · 2021-09-30
> > > **Good idea**
> > >
> > > Thank you for your suggestion. That is a nice idea and we will definitely try it, even though it will not make it in time for the discussion deadline.

---

### Official Review · Reviewer_1Kap · 2021-09-18
**A Fine-Grained Dataset of Architectural Styles with Real-World Challenges**

**Rating:** 6
**Confidence:** 3
**Clarity:** Paper is well written.

**Strengths:**

1). The proposed architectural styles dataset is novel and practical, which covers different types of images and labels compared to prior datasets.
2) The paper provides detailed statistical information.

**Weaknesses:**

1)  Limited takeaways. The authors claim that there are numerous real-world challenges: L456 fine-grained distinctions between classes based on subtle visual features, a highly imbalanced class distribution, but do not provide detailed experiment and insight to prove the related contribution to prior dataset or methods. Some things the authors could have included, are e.g. discussion of dataset difficulty (prior vs. theirs for fine-grained distinctions between classes tasks) as well as to show that the same model trained on the proposed large and the existing datasets. This is also something to do for the Imbalanced Class Distribution challenges.
2) Contribution/Comparison to prior datasets. It remains unclear whether the proposed dataset is indeed more challenging. I know that there are few prior works for architectural styles, and it's not the authors’ fault. But what is the main difficulty for the current hierarchy classification, imbalance class, fine-grained distinctions between classes, and what is the main solution for these challenges? Whether these methods are effective for the proposed dataset?
3) Chosen methods. The authors only provide one simple baseline with ResNet-50. It's hard to convince for so simple experiment setting without new insight and analysis.

**Additional Feedback:**

None

**Correctness:**

Lacking the convincing experiment, and the comparison to prior dataset or methods.

**Documentation:**

Yes

**Ethics:**

Yes

**Relation To Prior Work:**

No, there are many related previous works that need to discuss and compare, e.g., some works concerning class hierarchy, imbalanced class, fine-grained distinctions datasets, and methods. This could further refine the paper, providing deeper levels of analysis for some challenges and try to overcome them.

**Summary And Contributions:**

The authors introduce a novel dataset for architectural style classification, consisting of 9,485 images of church buildings, which can be used for fine-grained distinctions between classes based on subtle visual features, a highly imbalanced class distribution, a high variance of viewpoints.

---

> ### Author Response · Authors · 2021-09-29
> **Comparison with SOTA on related datasets**
>
> Thank you for your comments.
>
> We understand your wish for a comprehensive benchmark of methods for fine-grained classification and imbalanced class distributions across datasets. In this paper, in contrast, we focused on providing an thorough description of the novel WikiChurches dataset, its properties, and collection process.
>
> To integrate a comparison aspect anyway, we conducted additional experiments using [the current state-of-the-art method][2] on the fine-grained Stanford Cars dataset, according to [PapersWithCode][1]. The approach is based on sophisticated pre-training on the larger ImageNet-21k (as opposed to the often used ImageNet-1k) and fine-tuning on downstream tasks. We consider Stanford Cars as posing a comparable task to WikiChurches due to its small sample size, fine-grained distinctions between car models, and the fact that both cars and buildings should be sufficiently covered in ImageNet-21k used for pre-training.
>
> On the 196-class problem of Stanford Cars, this method achieves a quite high accuracy of 96.3%.
>
> We used the pre-trained TResNet-M model provided by the authors and followed [their instructions][3] for transfer learning. Since an increase of input resolution proved valuable in our existing experiments, we changed the input image size for the fine-tuning step to 448x448. Learning rate and weight decay were tuned on the train/val split using grid search.
>
> Without accounting for the class imbalance during training (because this was not done originally by the authors), this method obtains a balanced accuracy of 65.5% on WikiChurches-6 (averaged over 10 runs). This is comparable to our baseline of 65.9%, which used sub-sampling for training and a ResNet-50 architecture pre-trained on ImageNet-1k.
>
> Additional experiments with balanced training for this method could not be completed until the discussion deadline. Once ready, we will include both additional experiments in the camera-ready version.
>
> While both Stanford Cars and WikiChurches pose a fine-grained classification task of everyday objects (cars and church buildings) that should be sufficiently covered in ImageNet-21k, the discrepancy between 96.3% accuracy on Stanford Cars with 196 classes and 65.5% on WikiChurches-6 with only 6 classes indicates that the latter poses a much more difficult problem.
>
> [1]: https://paperswithcode.com/sota/fine-grained-image-classification-on-stanford?p=imagenet-21k-pretraining-for-the-masses
> [2]: https://arxiv.org/abs/2104.10972
> [3]: https://github.com/Alibaba-MIIL/ImageNet21K/blob/main/Transfer_learning.md

---

> > ### Comment · Reviewer_1Kap · 2021-09-29
> > **Reply to authors' response**
> >
> > Thank you very much for the response. The authors provided additional experiments using the sota method in the fine-grained Stanford Cars dataset, which achieve a dissatisfactory performance 65.5% on WikiChurches-6 with only 6 classes.
> >
> >  I appreciate the effort, but there still remain many concerns for me. The same as my original comments:
> >
> > What is the main difficulty/challenge for the current hierarchy classification, imbalance class, fine-grained distinctions on the proposed benchmark? I still think there needs more analysis and give new insights, not just show the performance with the sota method. For example, the authors can provide the performance of each class, and analyze which two are the most difficult case for fine-grained distinctions.
> >
> > In fact, I highly appreciate the contribution of the paper on dataset and annotation. But I still feel that the experiments(new insight and analysis), the Comparison to prior datasets need improvement.

---

> > > ### Author Response · Authors · 2021-09-29
> > > **Reply**
> > >
> > > We highly appreciate your continued engagement in helping us to improve the paper further. Of course, providing more interesting insights is in our very own interest. We are, however, not completely certain whether we understand your suggestion to "provide the performance of each class and analyze which two are the most case for fine-grained distinctions."
> > >
> > > We did provide the classification accuracy for each class in Fig. 5 and analyzed the main types of mistakes. This showed that the most fine-grained distinctions are between Renaissance and Baroque as well as between Gothic and Romanesque architecture. As we argued in Sec. 4.2, this is also plausible, because Baroque uses basically the same structural elements as Renaissance but exaggerates them. With regard to Gothic and Romanesque, one highly distinctive feature is the shapes of arches. While they are round in Romanesque architecture, they are pointed in Gothic style. This is a fine distinction which requires attention to detail.
> > >
> > > How can we improve this analysis to suit your demand better? We would highly appreciate if you could help us understand your suggestion better.

---

> > > > ### Comment · Reviewer_1Kap · 2021-09-30
> > > > **final comment**
> > > >
> > > > In fact, the ablation experiment like Fig. 5 with new insights and analysis is what I want to see. I will change my score to "6: Marginally above acceptance threshold". The related experiment and analysis for balanced training should be added in the revised version, just like the author's response. At the same time, I feel like I would like to see the full updated manuscript.

---

### Official Review · Reviewer_roSv · 2021-09-21
**A fine-grained church dataset**

**Rating:** 5
**Confidence:** 4
**Correctness:** The claims made in the submission are…
**Clarity:** The paper is well written.

**Strengths:**

1. Such a church dataset for fine-grained recognition does not exist before. This newly proposed dataset can benefit some specific fields such as fine-grained classification.
2. The dataset collection, cleaning, and annotation processes are described in detail and clearly.
3. The bounding box annotation for some images is interesting and useful for helping the model to learn fine-grained recognition.

**Weaknesses:**

1. As the authors mentioned in the paper, there are label noise and bias in the dataset, which might affect fine-grained recognition training.
2. The dataset can be potentially used in many tasks, but the authors only provide a simple baseline for image classification with resnet-50. Having a more comprehensive baseline on different tasks and different previous methods would give the readers a better understanding and insights of the dataset. For example, a benchmark of the performance of previous fine-grained classification and imbalanced classification methods on this dataset, the effect of imagenet pretraining on this dataset, etc.

**Additional Feedback:**

N/A.

**Documentation:**

There is sufficient detail on data collection and organization. The data is available online. There is no maintenance plan, ethical and responsible use provided.

**Ethics:**

No ethical concerns.

**Relation To Prior Work:**

It is clearly discussed.

**Summary And Contributions:**

This paper proposes a fine-grained church dataset for architectural style classification, sourced from Wikipedia. Some of the images are also annotated with bounding boxes of characteristic visual features. This dataset can be potentially used for different research problems,  fine-grained distinctions, imbalanced classification, and learning with a hierarchical organization of labels, where only some images are labeled at the most precise level. A classification baseline for this dataset is provided.

---

> ### Author Response · Authors · 2021-09-29
> **Further experiments**
>
> Thank you for your comments.
>
> We are not able to provide an extensive benchmark of various methods from different areas within the discussion period and the already scarce space available for the description of the dataset. On the other hand, we understand your wish for a more comprehensive baseline.
>
> Thus, we conducted additional experiments using [the current state-of-the-art method][2] on the fine-grained Stanford Cars dataset, according to [PapersWithCode][1]. The approach is based on sophisticated pre-training on the larger ImageNet-21k (as opposed to the often used ImageNet-1k) and fine-tuning on downstream tasks. We consider Stanford Cars as posing a comparable task to WikiChurches due to its small sample size, fine-grained distinctions between car models, and the fact that both cars and buildings should be sufficiently covered in ImageNet-21k used for pre-training.
>
> We used the pre-trained TResNet-M model provided by the authors and followed [their instructions][3] for transfer learning. Since an increase of input resolution proved valuable in our existing experiments, we changed the input image size for the fine-tuning step to 448x448. Learning rate and weight decay were tuned on the train/val split using grid search.
>
> Without accounting for the class imbalance during training, this method obtains a balanced accuracy of 65.5% (averaged over 10 runs). This is comparable to our baseline of 65.9%, which used sub-sampling for training and a ResNet-50 architecture pre-trained on ImageNet-1k.
>
> Additional experiments with balanced training for this method could not be completed until the discussion deadline. Once ready, we will include both additional experiments in the camera-ready version.
>
> [1]: https://paperswithcode.com/sota/fine-grained-image-classification-on-stanford?p=imagenet-21k-pretraining-for-the-masses
> [2]: https://arxiv.org/abs/2104.10972
> [3]: https://github.com/Alibaba-MIIL/ImageNet21K/blob/main/Transfer_learning.md

---

> ### Comment · Reviewer_roSv · 2021-10-04
> **Thank authors for the rebuttal**
>
> Thank authors for the rebuttal! I still think this is a borderline paper and the significance of a fine-grained church dataset with classification benchmark is not high enough. So I would keep my original rating of borderline.

---

### Decision · Program_Chairs · 2021-10-09

**Decision:**

Accept

**Comment:**

The paper introduces a new dataset of ~9.5k images of churches annotated with fine-grained and hierarchical labels of architectural style. Reviewers felt that the dataset was novel and well-described, and would be an interesting testbed for fine-grained recognition. The primary weakness mentioned by reviewers was that the paper does not provide sufficient baseline classification results on the dataset. The authors provided additional results in their response to Reviewer roSv. In addition, the paper does not justify the utility of the bounding box annotations. On the whole, the AC feels that the benefits of this dataset for fine-grained classification outweigh the weaknesses. Congratulations on having your paper accepted to the NeurIPS 2021 Datasets & Benchmarks Track! The authors are encouraged to take the feedback from reviewers into account when preparing the final version of the paper.